# Meeting the UN's Sustainable Development Goals in the Decarbonization Agenda: A Case of Russian Oil and Gas Companies

Nataliya Titova [1,*], Alina Cherepovitsyna [1] and Tatiana Guseva [2]

1 Luzin Institute for Economic Studies—Subdivision of the Federal Research Centre, Kola Science Centre of the Russian Academy of Sciences, 184209 Apatity, Russia; a.ilinova@ksc.ru
2 Research Institute "Environmental Industrial Policy Centre", 42 Olimpijskij Prospect, 141006 Mytishchi, Russia; t.guseva@eipc.center
* Correspondence: n.titova@ksc.ru; Tel.: +7-9532177326

**Abstract:** Being key players in providing sustainable energy on a global scale, oil and gas (O&G) companies can contribute to achieving the UN's Sustainable Development Goals SDG 7 (Affordable and Clean Energy) and SDG 13 (Climate Action). This paper focuses on Russian O&G companies and presents an analysis of their contribution to these SDGs in the context of today's decarbonization agenda. The study is based on a content analysis of their corporate sustainability reports and has produced three results. First, we analyzed the key strategic goals of Russian O&G companies. Second, we identified the correspondence between the progress towards the SDGs declared in their sustainability reports and the UN's SDG indicators. Third, we analyzed the contributions of Russian O&G companies to SDGs 7 and 13. As a result of the study, recommendations were formulated to introduce practical tools aimed at increasing the contribution of Russian O&G companies to sustainable development. The article discusses problems in corporate sustainability reporting of Russian O&G companies. The paper also seeks to expand the existing literature on the contribution of the Russian O&G sector to providing sustainable energy and accelerating the energy transition in line with the decarbonization agenda.

**Keywords:** sustainable development; sustainable development goals; SDGs; oil and gas companies; climate change; decarbonization; SDG contributions

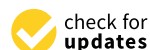



## 1. Introduction

Sustainable development ideas have been actively incorporated into science, business, and our daily lives globally. Today, sustainable development (SD) is understood as a holistic approach to achieving global wellbeing; 17 goals (SDGs) and 169 targets were set by the UN's 2030 Agenda for Sustainable Development [1]. These goals and targets aim at solving a number of issues, including reducing economic inequality, increasing social wellbeing, improving the environment, and developing innovation to preserve and improve the state of the planet for future generations. It is recognized that the largest companies in resource-intensive sectors of the economy have a crucial role to play in ensuring a contribution to the achievement of the SDGs, as they have the greatest degree of responsibility for using the planet's human and natural resources [2].

The influence of the energy sector on the implementation of the 2030 Agenda is significant; its performance has a direct impact on achieving SDG 7 (Affordable and Clean Energy), and its development plays a crucial role in achieving all other SDGs as it ensures that all the necessary transformations are made on the path to SD [3]. Meanwhile, the energy sector causes significant environmental impacts and contributes to climate change, making it difficult to achieve SDG 13 (Climate Action) [4]. In 2021, global carbon emissions

amounted to 36.3 billion tons [5], with 93% of emissions originating from the energy sector [6].

Therefore, energy companies should operate in such a way that need to minimize their negative impact on the climate in order to contribute to the SDGs. The energy sector should strive for carbon neutrality, even though this is a challenging objective. Decarbonization has become a trend along which companies plan their strategies within the SD concept [7]. It involves the implementation of low-carbon technologies by the energy sector to significantly decrease greenhouse gas emissions [8]. Generally, energy transition entails a shift away from traditional energy sources such as oil, natural gas, and coal in favor of alternative ones, including solar, wind, nuclear, and hydrogen energy.

The oil and gas (O&G) industry, which accounts for 52% of global energy consumption, plays a key role in the SD of the energy sector [9]. O&G companies are facing serious challenges in the context of an increasingly urgent climate agenda [10]. The Oil and Gas Climate Initiative (OGCI) plays a key role in developing decarbonization policies and achieving the SDGs [11]. O&G companies around the world that follow the OGCI and support the goals of the Paris Agreement aim to achieve carbon neutrality [12]. As it is necessary to dramatically reduce greenhouse gas (GHG) emissions, O&G companies activate decarbonization processes, which makes a significant contribution to the energy transition. To ensure sustainability, O&G companies are switching to renewables, working on their strategic flexibility and their processes and products with reduced GHG emissions [13].

Being the world's leading supplier of fossil fuels and petroleum products, Russia is also one of the largest emitters of greenhouse gases, accounting for about 4.7% of global GHG emissions in 2021 [14]. However, the government declares that it is actively integrating the SDGs into its domestic policies [15]. The Russian Federation's national projects are directly or indirectly related to most of the 169 SD targets [16]. Given the significant role of the Russian oil and gas sector in the global energy landscape and its fundamental position in the Russian economy, the experience of Russian O&G companies regarding SD, energy transition, and decarbonization strategies is particularly noteworthy.

In this context, decarbonizing the industry assumes critical importance in implementing the concept of SD under the Paris Agreement [17–19]. Consequently, the way O&G companies balance their contributions to SDG 7 (Affordable and Clean Energy) and SDG 13 (Climate Action) plays a major role in SD and energy transition [20]. As the O&G sector undergoes decarbonization, it becomes imperative to ensure growth in energy production while simultaneously reducing greenhouse gas emissions. In doing so, O&G companies contribute to SD and expedite the realization of a low-carbon economy.

According to the Oil and Gas Climate Initiative [11], O&G companies disclose information about their responsibilities in the field of SD using non-financial reports [21]. Various aspects of information disclosure in these non-financial reports reflect the requirements of several international organizations and their initiatives, including the Carbon Disclosure Project (CDP) [22], the Sustainability Accounting Standards Board (SASB) [23], the Task Force on Climate-Related Financial Disclosures (TCFD) [24], the United Nations Guiding Principles (UNGP Reporting Framework) [25], the International Petroleum Industry Environmental Conservation Association (IPIECA) [26], the Global Reporting Initiative (GRI) [27], the concept of Public Non-Financial Reporting by the Russian Union of Industrialists and Entrepreneurs [28], and others. Sustainability reports are the most common form of non-financial reporting, with the standards set by the GRI [27]. Sustainability reports serve as a tool for companies to disclose to stakeholders their stance on climate change responsibility [29].

Several studies have been conducted regarding the impact of information disclosure in non-financial reporting on the financial performance of O&G companies. Companies in this sector that fail to disclose climate risks in their sustainability reports and neglect to outline their strategies for transitioning to low-carbon energy sources may face limited access to debt financing [30], while reporting about climate change mitigation efforts positively influences their market capitalization [31]. An analysis of Russia's SD practices has revealed

that Russian companies are improving their sustainability reports to manage risks and improve financial stability. It should be noted that political factors have a major influence on capitalization and the development of carbon neutrality measures in the Russian O&G sector, as highlighted by E. Vetoshkina et al. [32]. We can conclude that the groundwork has been established for further studies into the relationship between SD reporting and the financial positions of Russian O&G companies both domestically and abroad.

The question of how O&G companies can make a significant contribution to achieving the SDGs is being actively discussed in today's scientific and business literature [33,34]. According to IPIECA, O&G companies can contribute to every SDG by integrating it into their core business operations and by identifying opportunities for collaboration with other stakeholders and leveraging experiences and resources in support of the goal [35,36]. The standard GRI 11: Oil and Gas Sector 2021 clearly states that O&G companies can make the most significant contribution to SDG 7 and SDG 13 [37].

A study by F. M. M. G. Borges et al. suggests that globally, O&G companies can make the greatest contribution to achieving SDG 7 (Affordable and Clean Energy), SDG 11 (Sustainable Cities and Communities), SDG 9 (Industry, Innovation, and Infrastructure), and SDG 12 (Responsible Consumption and Production). The authors substantiate the choice of these SDGs by explaining that cities account for 80% of global energy consumption, leading to a negative environmental impact that is proportionally big and goes against SDG 7 and SDG 11 [38]. To reduce this impact, O&G companies must develop resource efficient and environmentally sound technologies (which correlates with SDG 9 and SDG 12), which requires a significant amount of investment. Based on a sample of reports from the largest companies in the Latin American O&G sector, the study found that their contribution to achieving SDG 9 and SDG 12 is not enough. However, the study does not explore the contribution of oil and gas companies to SDG 13 or the energy sector's climate change impacts.

Another study on the sustainability initiatives of an Italian monopoly in the energy sector revealed that the activities performed in the field of sustainability are related to SDGs 12, 11, 9, and 15. The study also revealed that the SDGs can be considered from two perspectives: as a tool for assessing the company's operational efficiency and as an opportunity for the company to make a significant contribution at the macro level to the achievement of the 2030 Agenda goals, benefiting many stakeholders [39].

The issue of whether the sustainable practices implemented by O&G companies actually help to achieve the SDGs is also highlighted in A. Okeke's research, which shows that the choice of priority SDGs is influenced by regional and country-specific factors, resulting in differences between the Asian, European, and American models of SD for O&G companies [40]. The researcher classified sustainable practices based on the Triple Bottom Line (TBL) approach, identifying social, environmental, and economic dimensions taken to make a contribution to SD [41].

According to the study's findings, the Asian model of SD in O&G companies demonstrates sufficient strategic flexibility but a weak commitment to complying with the international information disclosure standards. Additionally, companies in both Asia and Europe express interest in developing low-carbon technologies and making an accelerated energy transition.

European O&G companies have the largest number of ongoing sustainable practices. Their strategies focus on reducing emissions and increasing the value added to their products. Their reports show a high level of information disclosure on socially beneficial activities, such as ensuring social guarantees and safe working conditions for employees. Unlike Asian companies, which only demonstrate their commitment to active decarbonization, reports from European companies contain information about subsequent investments in the development of low-carbon technologies and their implementation in the production process.

American companies tend to take a conservative stance towards low-carbon research and are less likely to invest in projects associated with renewables. Additionally, they

are characterized by a lower degree of social sustainability reporting compared to their counterparts in Asia and Europe.

To sum up the above, ensuring the SD of the energy sector is a demanding challenge, and whether it will be met depends on major O&G companies in every country around the world. In each country, O&G companies make their unique contributions to the SDGs, which necessitates further research in order to identify areas for improvement. Our literature review on the topic shows that there are studies on the SD strategies of O&G companies in Asia, America, and Europe.

However, the experience of O&G companies in Russia has not yet been studied thoroughly enough. There is some research available on the SD of the Russian O&G sector, although it is limited.

Among the studies in this domain are those devoted to factors contributing to better sustainability reporting [42–44] and the influence of non-financial reporting on a company's market capitalization [45]. There are also decarbonization studies that focus on aspects such as the O&G sector's strategic goals and development priorities, the potential for the use of low-carbon technologies, and the strategy of the Russian energy industry in the context of climate scenarios for the development of the global energy sector [46–48]. However, we have not found studies that analyze the real contribution to SD made by Russian O&G companies in the context of global climate change. This study aims to fill this research gap. It focuses on the largest Russian O&G companies, as they are vital elements of the Russian energy sector, by studying their real contribution to SDG 7 and SDG 13 as the most important SDGs in the context of energy transition, global climate change, and the decarbonization trend within the O&G industry. The article addresses the following research questions:

RQ1: Are SDG 7 and SDG 13 given priority by Russian O&G companies?

RQ2: What are the strategic goals of Russian O&G companies concerning the achievement of SDG 7 and SDG 13?

RQ3: What indicators do Russian oil and gas companies use to illustrate their contribution to SDG 7 and SDG 13 in the face of global climate change?

RQ4: What contribution to SD do Russian O&G companies demonstrate in the processes of decarbonization and energy transition?

In contrast to prior studies, this research aims to identify the specific contribution to SD made by Russian O&G companies while recognizing the limitations they encounter in compiling sustainability reports. It evaluates the extent to which Russian O&G companies contribute to the SDGs and formulates recommendations, expanding the existing literature on the potential that these companies have in terms of SDG achievement.

## 2. Materials and Methods

The literature search was conducted between July 2022 and May 2023 using the Science Direct, Scopus, and Web of Science databases. Its results helped us formulate the research questions and design our research algorithm presented in Figure 1.

In the next research step, an initial sample was selected from the largest O&G companies in Russia based on the Energy Sector of Russia 2020 data collection by the Analytical Center for the Government of the Russian Federation [49]. Since the data collection covers the period before 2019, we also used official annual financial reports. Table 1 presents the data on production volumes, revenue, and net profit for 2021.

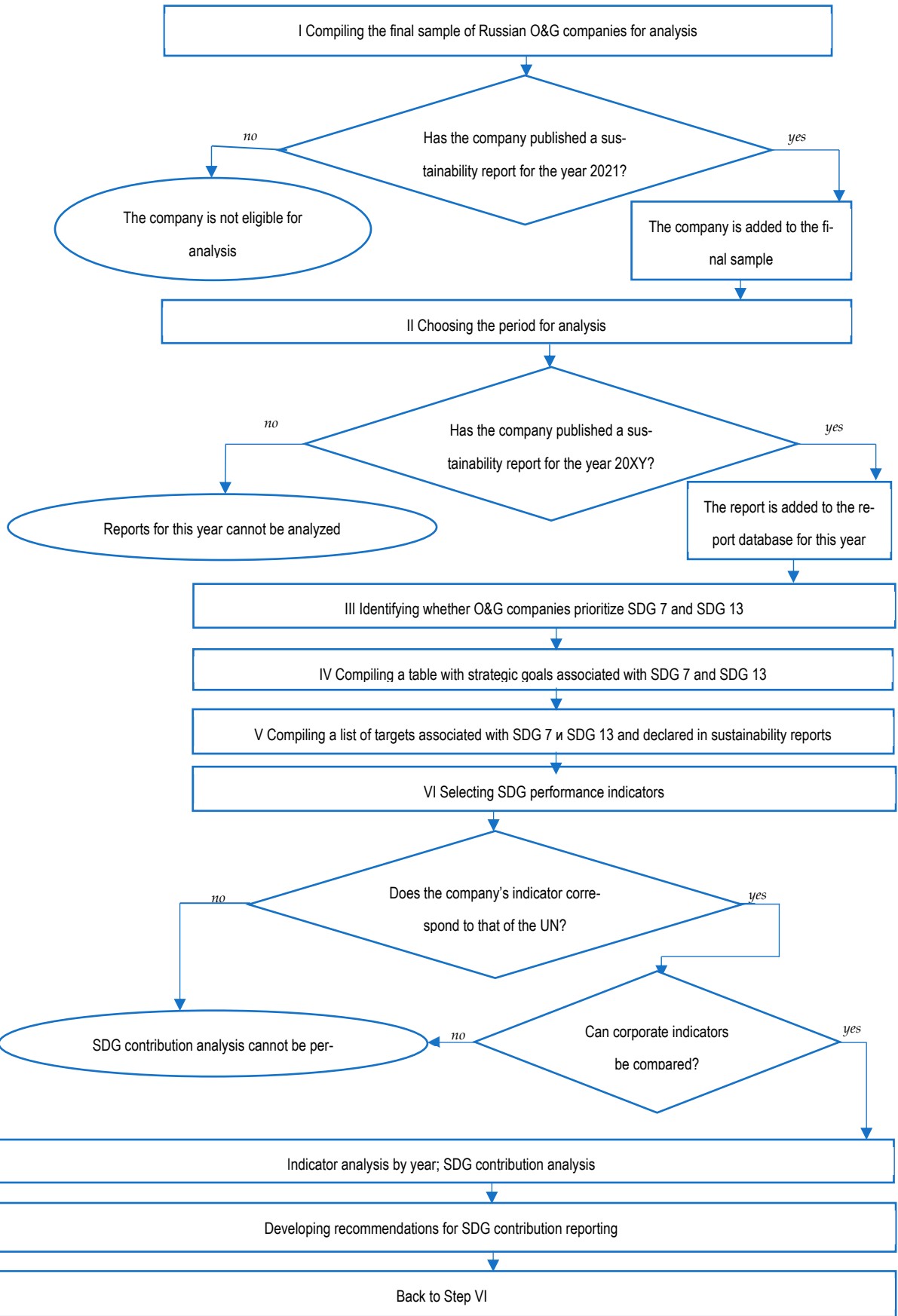

**Figure 1.** The structure of the research. Source: compiled by the authors.

**Table 1.** A sample of Russian O&G companies and their key indicators (2021).

| Company | Specialization | Oil Production, mln t | Gas Production, bcm | Revenue, bn RUB | Net Profit, bn RUB | Sustainability Report Availability for 2021 |
|---------|----------------|----------------------|---------------------|-----------------|--------------------|---------------------------------------------|
| Rosneft | Oil/gas | 192.1 | 64.7 | 8761.0 | 1012.0 | available |
| Lukoil | Oil/gas | 81.2 | 32.2 | 9435.1 | 775.5 | available |
| Gazprom Neft | Oil/gas | 62.2 | 47.9 | 3086.4 | 519.4 | not available |
| Surgutneftegas | Oil/gas | 55.5 | 9.1 | 1888.3 | 513.2 | not available |
| Tatneft | Oil | 27.8 | - | 1265.4 | 198.9 | available |
| Gazprom | Gas | - | 515.6 | 10,241.4 | 2684.5 | available |
| Novatek | Gas | - | 79.9 | 1156.7 | 451.6 | available |

Source: compiled by the authors, data from [50–60].

Then, a final sample was compiled based on whether the company had published a sustainability report for 2021, prepared in accordance with the GRI 102 standards [61], the IPIECA/API guidelines for voluntary reporting in the field of SD in the O&G sector [62], and the principles of the UN Global Compact [63]. All of the companies on the list have sections on their websites dedicated to sustainability reporting. However, Gazprom Neft did not publish its 2021 sustainability report, and the Sustainability section of its website was still under development. Surgutneftegaz published an environmental report, which differs in its structure from the standards accepted in the international community. Consequently, only Rosneft, Gazprom, Lukoil, Novatek, and Tatneft made it to the final sample.

In order to assess the contribution of Russian O&G companies to the SDGs, the data must be comparable, which means that the corporate data must be analyzed over the same period for the companies in the sample. In the next research step, a database was formed from sustainability reports for several periods. The main requirement was the availability of a sustainability report for each of the companies in the sample for a given period. Table 2 presents the data on the years when the companies published their sustainability reports.

**Table 2.** Russian O&G companies and sustainability reporting: Reporting years.

| Company in the Final Sample | Sustainability Reporting Years |
|-----------------------------|-------------------------------|
| Rosneft | 2006–2021 |
| Gazprom | 2008–2021 |
| Lukoil | 2020–2021 |
| Novatek | 2004–2021 |
| Tatneft | 2015–2021 |

Source: compiled by the authors, data from [51,53,56,58,60,64–73].

As the table shows, Novatek, Rosneft, and Gazprom started their sustainability reporting quite early (in 2004, 2006, and 2008, respectively). However, the early versions of their sustainability reports do not contain indicators corresponding to those established by the UN, which means they cannot be used for the purpose of the study. The period from 2019 to 2021 was chosen for analysis, as it is during these years that the companies provided the most detailed data on their sustainability indicators.

The sustainability reports were studied using the content analysis method. They can be decomposed into three major sections. Section 1 includes a description of the general issues regarding the concept and strategy of the company's SD management system (slogan; mission; strategic goal; message from the Management Board; and SD concept and principles). Section 2 contains information about the company's SDG contribution. Section 3 discusses sustainability measures in the relevant environmental, social, or economic areas. The content analysis of Section 1 identified whether O&G companies prioritize SDG 7 and SDG 13 (RQ 1). In this research step, charts and tables were also used.

Using the content analysis method, a search was made for the strategic goals that the company sets regarding SDG 7 and SDG 13 (RQ 2). A comparison was made between the strategic goals and the SDG targets.

The SDG contribution section, which is present in all of the sustainability reports analyzed, served as a foundation for identifying the sustainability indicators used by the companies (RQ 3). The data published in this section allows for drawing a conclusion about the progress in achieving the SDG targets 7 and 13 (RQ 4).

In the process of the research and content analysis, we encountered a number of difficulties, including the fact that some of the companies in the sample (Rosneft and Tatneft specifically) do not mention particular SDG targets in their reports in the sections devoted to the company's SDG contribution. To overcome this problem, we compared the indicators across the sample based on their essence. This research step provided for making a conclusion as to the extent to which the indicators used by Russian O&G companies correspond to those proposed by the UN for assessing SDG contributions. The sustainability reports were then scanned for the tools that Russian O&G companies used to make progress.

We analyzed SDG contributions across the Russian O&G sector only for those indicators that comply with the UN's methodology. We shall note that an SDG contribution cannot be analyzed if a company included in the sample does not report on a particular indicator, disabling comparisons. However, it is possible to formulate recommendations so that the SDG contribution can be monitored and evaluated.

## 3. Results

### 3.1. Russian Oil and Gas Companies in the Face of Climate Change: Identifying Priority SDGs and the Strategic Focus

By studying sustainability reports for the years 2019 to 2021, we identified the SDGs that the companies prioritized over this period. The results are shown in Table 3.

**Table 3.** Russian oil and gas companies and their priority SDGs.

| Company | Year | Sustainability Report Page | SDG |
|---|---|---|---|
| Rosneft | 2019 | 6 | 3, 7, 8, 13, 17 |
| | 2020 | 6 | 3, 7, 8, 13, 17 |
| | 2021 | 4–5 | 3, 7, 8, 13, 17 |
| Gazprom | 2019 | - | No priority SDGs |
| | 2020 | - | No priority SDGs |
| | 2021 | 48 | 3, 4, 7, 8, 9, 13 |
| Lukoil | 2019 | - | 4, 5, 6, 7, 8, 9, 12, 13, 14, 15, 17 |
| | 2020 | - | Sustainability report not publicly available |
| | 2021 | 4 | 4, 5, 6, 7, 8, 9, 12, 13, 14, 15, 17 |
| Novatek | 2019 | 30–32 | 3, 4, 7, 8, 13 |
| | 2020 | 32 | 3, 4, 7, 8, 13 |
| | 2021 | 4 | 3, 4, 7, 8, 13 |
| Tatneft | 2019 | - | No priority SDGs |
| | 2020 | 106 | 3, 4, 6, 7, 9, 11, 12, 13, 15, 17 |
| | 2021 | 12–13 | 3, 4, 6, 7, 9, 11, 12, 13, 15 |

Source: compiled by the authors, data from [51,53,56,58,60,64–73].

All the companies in the final sample chose SDG 7 and SDG 13 as their priority goals for the year 2021. Thus, they can be classified as the key SDGs for the Russian O&G sector

and used as a foundation for assessing corporate SDG contributions. Such assessments are in line with the climate agenda of O&G companies and the processes of decarbonization and energy transition, which are the focal point of this study. Table 4 provides further insight into the targets and indicators proposed by the UN for achieving the SDGs.

**Table 4.** Targets and indicators for achieving SDG 7 (Affordable and Clean Energy) and SDG 13 (Climate Action) according to the UN.

| SDG Targets | SDG Indicators |
|---|---|
| **Goal 7. Ensure access to affordable, reliable, sustainable and modern energy for all** | |
| 7.1 By 2030, ensure universal access to affordable, reliable and modern energy services | 7.1.1 Proportion of population with access to electricity |
| | 7.1.2 Proportion of population with primary reliance on clean fuels and technology |
| 7.2 By 2030, increase substantially the share of renewable energy in the global energy mix | 7.2.1 Renewable energy share in the total final energy consumption |
| 7.3 By 2030, double the global rate of improvement in energy efficiency | 7.3.1 Energy intensity measured in terms of primary energy and GDP |
| 7.a By 2030, enhance international cooperation to facilitate access to clean energy research and technology, including renewable energy, energy efficiency and advanced and cleaner fossil-fuel technology, and promote investment in energy infrastructure and clean energy technology | 7.a.1 International financial flows to developing countries in support of clean energy research and development and renewable energy production, including in hybrid systems |
| 7.b By 2030, expand infrastructure and upgrade technology for supplying modern and sustainable energy services for all in developing countries, in particular developed countries, small island developing States and landlocked developing countries, in accordance with their respective programs of support | 7.b.1 Installed renewable energy-generating capacity in developing countries (in watts per capita) |
| **Goal 13. Take urgent action to combat climate change and its impacts [1]** | |
| 13.1 Strengthen resilience and adaptive capacity to climate-related hazards and natural disasters in all countries | 13.1.1 Number of deaths, missing persons and directly affected persons attributed to disasters per 100,000 population |
| | 13.1.2 Number of countries that adopt and implement national disaster risk reduction strategies in line with the Sendai Framework for Disaster Risk Reduction 2015–2030 |
| | 13.1.3 Proportion of local governments that adopt and implement local disaster risk reduction strategies in line with national disaster risk reduction strategies |
| 13.2 Integrate climate change measures into national policies, strategies and planning | 13.2.1 Number of countries with nationally determined contributions, long-term strategies, national adaptation plans and adaptation communications, as reported to the secretariat of the United Nations Framework Convention on Climate Change |
| | 13.2.2 Total greenhouse gas emissions per year |
| 13.3 Improve education, awareness-raising and human and institutional capacity on climate change mitigation, adaptation, impact reduction and early warning | 13.3.1 Extent to which (i) global citizenship education and (ii) education for sustainable development are mainstreamed in (a) national education policies; (b) curricula; (c) teacher education; and (d) student assessment |

**Table 4.** *Cont.*

| SDG Targets | SDG Indicators |
|---|---|
| 13.a Implement the commitment undertaken by developed-country parties to the United Nations Framework Convention on Climate Change to a goal of mobilizing jointly $100 billion annually by 2020 from all sources to address the needs of developing countries in the context of meaningful mitigation actions and transparency on implementation and fully operationalize the Green Climate Fund through its capitalization as soon as possible | 13.a.1 Amounts provided and mobilized in United States dollars per year in relation to the continued existing collective mobilization goal of the $100 billion commitment through to 2025 |
| 13.b Promote mechanisms for raising capacity for effective climate change-related planning and management in least developed countries and small island developing States, including focusing on women, youth and local and marginalized communities | 13.b.1 Number of least developed countries and small island developing States with nationally determined contributions, long-term strategies, national adaptation plans and adaptation communications, as reported to the secretariat of the United Nations Framework Convention on Climate Change |

Source: [74]. [1] Acknowledging that the United Nations Framework Convention on Climate Change is the primary international, intergovernmental forum for negotiating the global response to climate change.

Next, we will identify the strategic goals set by Russian O&G companies in the pursuit of SDG 7 and SDG 13.

### 3.2. Russian O&G Companies' Strategic Goals and Priority SDGs: Alignment Assessment

As O&G companies operate in the energy sector, SDG 7 is particularly relevant for them, which is illustrated by their reports. Table 5 compares the companies' strategic goals and SDG 7 targets.

As can be seen from the table, not all the companies under the study's set strategic goals in the SDG 7 domain that can be measured. Moreover, not all SDG 7 targets are covered by strategic goals. For example, SDG target 7.1 is only covered by Novatek, with the measurable strategic goals of increasing liquefied natural gas (LNG) production by up to 70 million tons per year by 2030 and providing LNG to remote consumers by 2025. Other companies' sustainability reports lack clear indicators for the provision of affordable energy. It should be noted that energy in Russia is affordable even to people on lower incomes, and prices lay well below those characteristic of Europe and the US. At the same time, according to the Russian Energy Strategy, all energy sector companies have to develop techniques aimed at the minimization of negative environmental impacts and the gradual transfer to low-carbon energy generation.

SDG target 7.2 is covered by Lukoil, Novatek, and Tatneft. While being considered leaders in the Russian oil and gas sector, Rosneft and Gazprom do not set any renewable energy goals. Lukoil sets an ambitious goal of transitioning to 100% energy consumption from renewable energy sources, although the timeframe for achieving this goal is not specified. Tatneft, on the other hand, presents clear quantitative criteria in its report, aiming to generate renewable energy of up to 426 MW by 2030 and 900 MW by 2050 [56], p. 14.

Strategic goals corresponding to SDG target 7.3 are defined by Rosneft, Gazprom, and Tatneft. However, clear quantitative criteria to assess their achievement are only declared in the sustainability reports of Gazprom and Tatneft. Rosneft's goal of improving energy efficiency is vague and lacks specific indicators and deadlines.

**Table 5.** Strategic goals reported by Russian O&G companies and SDG 7 targets.

| Company | Strategic Goal | SDG Target |
|---|---|---|
| Rosneft, [51], p. 4 | Rosneft takes steps to *improve energy efficiency* in all of its business activities and recognizes leadership in innovation as a key development driver.<br>The Company recognizes its role and *responsibility in providing timely and reliable energy supplies to consumers* (including in emerging markets) on equal terms and at competitive prices. | 7.1 By 2030, ensure universal access to affordable, reliable and modern energy services<br>7.3 By 2030, double the global rate of improvement in energy efficiency |
| Gazprom, [58], p. 4 | *Reduction in specific consumption of fuel and energy* for trunkline needs (under comparable operating conditions) by 12% by 2024 and by 17% by 2035 versus the 2018 level. | 7.3 By 2030, double the global rate of improvement in energy efficiency |
| Lukoil, [53], p. 44 | *Shifting to 100% renewable energy* and removing GHG emissions and radioactive waste materials from its supply chain. | 7.2 By 2030, increase substantially the share of renewable energy in the global energy mix |
| Novatek, [60], p. 16 | *To increase LNG production* from Company projects up to 70 million tons per year by 2030;<br>To supply LNG to consumers in areas remote from existing gas transmission infrastructure by 2025;<br>*To expand the use of renewables.* | 7.1 By 2030, ensure universal access to affordable, reliable and modern energy services<br>7.2 By 2030, increase substantially the share of renewable energy in the global energy mix |
| Tatneft, [52], p. 14 | *Energy generation (share increase) using RES* to 426 MW by 2030 and 900 MW by 2050;<br>*Increase in energy efficiency and energy saving* to a level of at least 2.2% of the actual consumption of fuel and energy resources in the previous year. | 7.2 By 2030, increase substantially the share of renewable energy in the global energy mix<br>7.3 By 2030, double the global rate of improvement in energy efficiency |

Source: compiled by the authors, data from [51,53,56,58,60,74].

SDG 13 (Climate Action)

Next, let us examine the strategic goals set by Russian O&G companies within the SDG 13 domain (see Table 6). Each of the companies in the sample has a long-term strategic goal regarding climate issues. Oil production and refining companies, such as Rosneft, Lukoil, and Tatneft, aim for carbon neutrality by 2050. Gas production and processing companies, however, do not set such goals, and their maximum planning horizon is 2030, with the objective of reducing specific greenhouse gas emissions.

The strategic goals analyzed align with SDG target 13.2 ("Integrate climate change measures into national policies, strategies and planning") and Indicator 13.2.2 (total greenhouse gas emissions per year). Russian O&G companies did not set goals that would align with SDG target 13.3 ("Build knowledge and capacity to meet climate change") or 13.1 ("Strengthen resilience and adaptive capacity to climate-related hazards and natural disasters in all countries").

**Table 6.** Strategic goals reported by Russian O&G companies and SDG 13 targets.

| Company | Strategic Goal Formulation | SDG Target |
|---------|----------------------------|------------|
| Rosneft, [51], p. 18 | - reduce absolute Scope 1 and 2 GHG emissions by 5% by 2025;<br>- reduce absolute Scope 1 and 2 GHG emissions by more than 25% by 2035;<br>- reduce methane intensity to below 0.2% by 2030;<br>- achieve carbon neutrality in terms of Scope 1 and 2 GHG emissions by 2050. | |
| Gazprom, [58], p. 76 | - 11.2% target reduction in specific GHG emissions by 2030;<br>- no less than 1.5% reduction specific GHG emissions in Scope 1;<br>- no less than 2.3% reduction in specific fuel and energy consumption for internal process needs and losses;<br>- 55.3 t of $CO_2$-eq/bcm ● km reduction in GHG emissions during natural gas transportation in terms of volumes of gas transported. | |
| Lukoil [53], p. 32 | - achieve net zero controlled emissions by 2050;<br>- decarbonization and adapting to climate change;<br>- reduce controlled GHG emissions (Scope 1 + Scope 2) by 20% compared to the 2017 level. | 13.2 Integrate climate change measures into national policies, strategies and planning |
| Novatek [60], p. 16 | - reduce GHG emissions per unit of production in the upstream segment by 6% from a 2019 baseline by 2030;<br>- reduce methane emission intensity by 4% from a 2019 baseline by 2030;<br>- reduce GHG emissions per ton of LNG produced by 5% from a 2019 baseline by 2030. | |
| Tatneft [56], p. 14 | - achieving carbon neutrality in 2050;<br>- reducing the intensity of greenhouse gas emissions. | |

Source: compiled by the authors, data from [51,53,56,58,60,74].

*3.3. Measuring Contribution to SD in the Context of Global Climate Change: Indicators Used by Russian O&G Companies*

3.3.1. SDG 7 (Affordable and Clean Energy)

The companies under study declare their contribution to SDG targets 7.1, 7.2, and 7.3, but not all of them provide the corresponding data either in the section on sustainability or elsewhere in their reports. An analysis of the SDG 7 indicators used by Russian O&G companies reveals some issues. First, there is an inconsistency between the proposed indicators and the UN's indicators. Second, there is an inconsistency between the corporate indicators and the formulations of the SDG targets (see Table 7).

To demonstrate their SDG target 7.1 contributions, Russian O&G companies disclose data such as the volumes of electricity sold and hydrogen produced (Gazprom, 2019–2021). They also include measures to improve their product quality management systems as part of their SDG target 7.1 contributions that can be divided into two groups. The measures of the first group focus on improving the environmental and performance characteristics of the energy sources produced. They include quality control laboratories, compliance with the ISO 9001 standard, unannounced audits, and adherence to the international environmental standards. The measures of the second group involve improving the quality and efficiency of the services provided to end users based on a customer satisfaction analysis. However, the indicators used to evaluate the effectiveness of these measures do not align with the UN's indicators, 7.1.1 and 7.1.2. Furthermore, since Russian O&G companies do not provide the data on the number of people consuming their electricity, it is not possible to evaluate their SDG 7.1 contributions.

**Table 7.** SDG 7 indicators used by Russian O&G companies.

| UN Targets and Indicators | Corporate Indicators Used in the Russian O&G Sector | Company; Available Years |
|---|---|---|
| 7.1 "By 2030, achieve universal access to affordable, reliable and modern energy supplies" | | |
| 7.1.1 Proportion of population with access to electricity 7.1.2 Proportion of population with primary reliance on clean fuels and technology | Gas sales, bcm | Gazprom, 2019–2021 [58,66,67] |
| | Electric and heat power sales, mln Gcal | Gazprom, 2019–2021 [58,66,67] |
| | Hydrogen output, thousand tons | Gazprom, 2019–2021 [58,66,67] |
| 7.2 By 2030, increase substantially the share of renewable energy in the global energy mix | | |
| 7.2.1 Renewable energy share in the total final energy consumption | RES development projects, million RUB | Lukoil, 2019–2021 [52,68,69] |
| | Employees trained, thousand people | Novatek, 2020–2021 [60,70] |
| 7.3 By 2030, double the global rate of improvement in energy efficiency | | |
| 7.3.1 Energy intensity measured in terms of primary energy and GDP | Electric power generation from renewable energy sources and secondary energy resources, mln MWh | Gazprom, 2019–2021 [58,66,67] |
| | Investments in renewable and secondary energy sources, billion RUB | Gazprom, 2019–2021 [58,66,67] |
| | Fuel and energy savings resulting from the implementation of energy saving programs, mln GJ | Gazprom, 2019–2021 [58,66,67]; Novatek, 2020–2021 [60,70]; Rosneft, 2019–2021 [51,64,65] |
| | Fuel and energy savings resulting from the implementation of energy saving programs, billion RUB | Rosneft, 2019–2021 [51,64,65] |
| | Savings in consumption volume under the energy efficiency program, % | Tatneft, 2021 [56] |
| | Expenditure associated with energy conservation programs, million RUB | Lukoil, 2019–2021 [52,68,69] |
| | Compliance with ISO 50001 (Energy Management Systems), % | Rosneft, 2021 [51,64,65] |

Source: compiled by the authors, data from [51,53,56,58,60,64–74].

O&G companies can make their SDG target 7.2 contributions by either consuming or generating renewable energy sources (RES). The former can be performed via the operation of wind and solar power systems. Among the companies under study, Lukoil is the only one that declares commercial power generation from RES. However, the company only discloses the total costs of its RES development projects, which have increased by a factor of 2.8 compared to 2019. Lukoil operates four hydropower plants, seven solar power plants, and one wind power plant [53,68,69].

Gazprom also declares its SDG target 7.2 contribution by reporting on its electric power generation from and investments in RES and secondary energy resources (mln MWh and bln RUB, respectively) [58]. However, the company classifies it as a SDG target 7.3 contribution. Such inconsistencies arise because Russian O&G companies consider RES as a means to contribute to the implementation of SDG 13 and other strategic goals aimed at reducing adverse environmental impacts. They see it as an integral step towards energy transition and decarbonization. Therefore, some companies report their RES progress as a contribution to achieving SDG 13.

As one of the key contributors to SDG target 7.2, Novatek reports switching from conventional energy to renewable energy sources. The Cryogas-Vysotsk LNG plant in the Leningrad Region has been using renewable energy since the beginning of 2022. The company conducted wind measurements to explore the potential for building a wind farm in the village of Sabetta on the Yamal Peninsula [60], p. 10.

Tatneft implements projects involving wind turbine construction, downhole power generation, and the introduction of pellet heating equipment and solar power plants at

their gas stations [56], p. 199. The company also operates a solar power plant and develops hydropower. However, in this study, renewable energy sources refer specifically to solar and wind energy, excluding energy generated from hydroelectric power plants.

Rosneft demonstrates the smallest SDG 7 contribution. The company has no ongoing projects in the field of renewable energy development, but it is piloting the "green office" model where all office electricity is generated using renewable energy sources [51], p. 92.

Contributing to SDG target 7.3 is considered by Russian O&G companies as "a lever to deliver against the GHG emissions reduction targets" [51], p. 92, allowing them "to consolidate the economic efficiency of operations" [58].

Content analysis of the sustainability reports identifies managerial, technological, and infrastructural methods for improving energy efficiency. Among these methods, those of the first group are the most widely used. Examples include well-developed management systems for improving energy efficiency. Gazprom, Novatek, Lukoil, and Tatneft declare in their sustainability reports that the key element of their energy management systems is the annually updated energy conservation program, within which short- and medium-term planning is carried out to ensure cuts in energy consumption to decrease direct GHG emissions (Tatneft, [56], p. 198; Novatek, [60], p. 64; Lukoil, [53], p. 37). Also, the companies are working on getting their energy management systems certified against the ISO 50001:2018 standard (Gazprom, [58], p. 91; Rosneft; [50], p. 47, Tatneft, [56], p. 198; Lukoil, [53], p. 17).

Technological methods for improving energy efficiency encompass the optimization of costs for lighting, power supply, and heating, as well as the application of energy-saving equipment and technologies. Companies declare the replacement of mercury and halogen lamps with energy-efficient LED lamps, the installation of automatic outdoor lighting control systems, and reactive power compensation to optimize energy consumption costs (Rosneft; [51], p. 47). Energy-saving equipment includes the use of third-generation energy-efficient pumping units (Tatneft, [56] p. 60), the replacement of asynchronous submersible motors with permanent magnet motors (Lukoil, [53], p. 37), the replacement of centrifugal compressors with electric ones, and the use of domestically produced steel pipes (Gazprom, [58], p. 22, p. 85). These tools can significantly reduce fuel consumption and greenhouse gas emissions by minimizing production losses.

Infrastructure methods involve upgrading production facilities. Tools in this category include the use of cogeneration technologies to improve energy efficiency and enhanced oil recovery techniques, as well as the reorganization of energy consumption patterns, which requires the substantial restructuring of production processes. Novatek, for example, declares the use of cogeneration technologies, with heat generation from secondary energy resources increasing by 20% to 2.4 million GJ in 2021, accounting for 69% of total heat consumption [60], p. 65. When discussing refining, Lukoil mentions "technical upgrade, optimization of production processes and distribution of energy flows and heat exchange between technological facilities" [53], p. 32. In oil production, the company uses energy-saving methods for enhanced oil recovery.

All of the above demonstrates that Russian O&G companies pay attention to energy efficiency issues, and each company has its own strategy for contributing to the achievement of SDG target 7.3. However, the information provided in the sustainability reports regarding the results of energy-saving strategies does not align with the UN's Indicator 7.3.1. The reports do not discuss primary energy consumption and energy intensity in their sections on sustainability, presenting this information elsewhere. Instead, companies report savings in fuel and energy costs or expenses associated with energy conservation programs (Lukoil, 2019–2021; Rosneft, 2019–2021) and the percentage reduction in energy consumption resulting from the implementation of energy-saving programs [51,53,64,65,68,69].

Some of the companies (Rosneft, Lukoil) emphasize that improving energy efficiency is one of the key measures for reducing GHG emissions [51,53,64,65,68,69]. Energy efficiency initiatives are cost-effective for these companies as they lead to reductions in operating costs. SDG target 7.3 contributions can be assessed using the energy intensity indicator proposed

by the UN. The companies under study have been disclosing their energy consumption since they started publishing their sustainability reports, but information on energy intensity indicators was not published until 2019, with some companies (Rosneft specifically) still not providing any data on energy intensity. Moreover, corporate energy intensity indicators differ in terms of the units of measurement (kg of reference fuel/thousand $m^3$ of natural gas; GJ/boe; GJ/t; kWh/thousand $m^3$ of natural gas; and kWh/t). Since this indicator is relative, it is not possible to aggregate the data and calculate the total energy intensity of companies in the Russian O&G sector.

To summarize the above, the only indicator that aligns with the UN's criteria is the share of renewable energy in energy consumption. By analyzing how this indicator has been changing over time, we can determine the contribution of Russian O&G companies to the achievement of SDG target 7.2. This analysis will be presented in the next section of this study.

3.3.2. SDG 13 (Climate Action)

While all the companies in the final sample prioritize SDG 13, their declared contributions are limited to Target 13.1. To identify their contributions to other SDG 13 targets, a content analysis of the full text of the sustainability reports is necessary. Table 8 presents the indicators declared by Russian O&G companies as their SDG 13 contribution measurements.

**Table 8.** SDG 13 indicators used by Russian O&G companies.

| UN Targets and Indicators | Corporate Indicators Used in the Russian O&G Sector | Company; Available Years |
|---|---|---|
| 13.1 Strengthen resilience and adaptive capacity to climate-related hazards and natural disasters in all countries | | |
| 13.1.1 Number of deaths, missing persons and directly affected persons attributed to disasters per 100,000 population<br>13.1.2 Number of countries that adopt and implement national disaster risk reduction strategies in line with the Sendai Framework for Disaster Risk Reduction 2015–2030<br>13.1.3 Proportion of local governments that adopt and implement local disaster risk reduction strategies in line with national disaster risk reduction strategies | Reduction in greenhouse gas emissions, % | Gazprom, 2019–2021 [58,66,67]<br>Novatek, 2020–2021 [60,70] |
| | Greenhouse gas emissions, mln t CO$_2$-eq | Gazprom, 2019–2021 [58,66,67]; Rosneft, 2019–2021 [51,64,65]; Tatneft, 2021 |
| | Reduction in GHG emissions per unit of production, % | Novatek, 2020–2021 [60,70] |
| | Reduction in methane emissions, % | Gazprom, 2019–2021 [58,66,67]; Rosneft, 2019–2021 [51,64,65] |
| | Current expenditures on air protection and climate change prevention, billion RUB | Gazprom, 2019–2021 [58,66,67]; Lukoil, 2019–2021 [53,68,69] |
| | Number of people trained in environmental programs, people | Gazprom, 2019–2021 [58,66,67] |
| | APG utilization, % | Novatek, 2020–2021 [60,70]; Tatneft, 2021 [56] |

Source: compiled by the authors, data from [51,53,56,58,60,64–74].

The most commonly reported indicator for demonstrating the Target 13.1 contribution among Russian O&G companies is greenhouse gas emissions, presented in both absolute and relative terms. Novatek also provides the data on the reduction in GHG emissions per unit of production, as this indicator is used by the company to measure its progress toward its strategic environmental goal [60,70,71]. Additionally, companies use indicators such as a reduction in methane emissions, associated petroleum gas (APG) utilization, and the number of staff trained in environmental programs. Lukoil and Gazprom disclose the costs related to air protection [53,58]. However, it is important to note that these indicators do not align with SDG target 13.1 indicators listed in Table 8. Instead, these indicators are more consistent with SDG target 13.2 and its associated indicator, the volume of greenhouse gas emissions.

Let us examine how Russian O&G companies make their contributions to SDG target 13.2 ("Integrate climate change measures into national policies, strategies and planning"). This challenge entails ensuring the alignment between the UN's SDGs, national targets for carbon footprint reduction, and the future goals of O&G companies. Some attempts to establish this

link between national goals and company contributions can be observed in the sustainability reports of Gazprom and Lukoil. These companies reflect their contributions to the development of National Environmental Programs, allowing for an assessment of their potential contribution to fulfilling the carbon reduction commitments made by the Russian Federation and tracking each company's contribution to the overall outcome [53,58]. However, other companies do not disclose their involvement in national environmental projects.

Considering the tools employed by O&G companies to contribute to the implementation of SDG 13, the following can be stated: the development of low-carbon technologies is viewed as a key instrument. For instance, Rosneft engages in cooperation with the China National Petroleum Corporation (CNPC) to develop carbon capture and storage technologies [50], p. 21.

Lukoil and Novatek are also taking steps towards collaborations in areas such as green energy, hydrogen production, and biofuels [53], p. 22; [60], p. 54. Novatek specifically focuses on low-carbon ammonia and hydrogen production. Lukoil and Rosneft are considering the potential of introducing projects for the production of new low-emission products, such as blue or green hydrogen, biofuels, and green aviation fuel, to reduce emissions in Scope 3. These projects are seen as key levers to achieve greenhouse gas emission reduction targets. Additionally, the companies analyze opportunities for their development in the context of climate change, including the production and sale of low-carbon energy products like biofuel and hydrogen [50], p. 21; [53], p. 37. In 2021, Novatek conducted feasibility studies in two areas: carbon dioxide capture and utilization, and the production, storage, transportation, and use of hydrogen and ammonia as low-carbon fuels [60], p. 39. However, it should be noted that these initiatives are still in the early stages, and significant progress is required before a full transition to low-carbon technologies can be achieved.

Among the mature initiatives for the implementation of low-carbon technologies, Gazprom has achieved major results in producing hydrogen from natural gas without $CO_2$ emissions by using plasma and molten metal for methane pyrolysis [58], p. 86. Their annual hydrogen output has reached 350,000 tons. In the reporting year, the company acquired two new patents for the utilization of low-pressure flare gases. The new technology splits low-pressure gas into liquid hydrocarbons and fuel gas, preventing the flaring of low-pressure gases. This technology enhances production efficiency, increases energy savings, and integrates quickly into the existing processes without requiring major changes [58], p. 97.

In 2021, Tatneft put into commercial operation units for hydrogen production, catalytic cracking, chemical water treatment, delayed coking, diesel fuel isodewaxing, and a gas fractionation unit [60], p. 36.

We can conclude that initiatives to introduce low-carbon technologies are still in the early stages. This is illustrated by the fact that the companies are conducting feasibility studies to assess the effectiveness of these technologies and searching for partners.

Regarding Target 13.3, the companies are focusing on training their staff and informing their stakeholders on climate compliance issues. Rosneft develops its in-house staff training programs in the fields of climate regulation and carbon management. Gazprom, Lukoil, and Novatek rely on the third-party courses on SD, while Tatneft conducts training courses on fostering an environmental outlook among the local population. However, these practical tools lack systematic implementation, as each company dictates its own standards, and there is a shortage of training programs for certified personnel. Consequently, there are no specific results reported that allow us to draw a definitive conclusion that Russian O&G companies are making a significant Target 13.3 contribution.

Hence, the contribution of Russian O&G companies to the achievement of SDG 13 can only be assessed via Target 13.2, which measures total annual greenhouse gas emissions. In the next section of this study, we will analyze trends in the achievement of corporate priority SDGs based on the UN indicators, starting from the earliest period for which the data is publicly available.

### 3.4. Russian O&G Companies and Their SDG Contributions

3.4.1. Corporate Contributions to SDG 7 (Affordable and Clean Energy)

The companies under study disclose the level of conventional energy consumed. They have also been reporting on renewable energy consumption since 2019 (though Lukoil started earlier—in 2017), which means their SDG target 7.2 contributions can be calculated from that year onwards. An exception is Rosneft, as the company currently does not provide information on renewable energy consumption. This lack of data can be attributed to the fact that Rosneft does not have strategic goals related to increasing the share of renewables in energy consumption.

The indicator for assessing SDG target 7.2 contributions can be calculated as the ratio of final renewable energy consumption to the total energy consumption, expressed as a percentage. In our calculations, we converted the units of measurement to ensure comparability. Russian O&G companies mainly use units such as thousand kWh and million GJ for measuring energy consumption.

The results are presented in Figure 2.

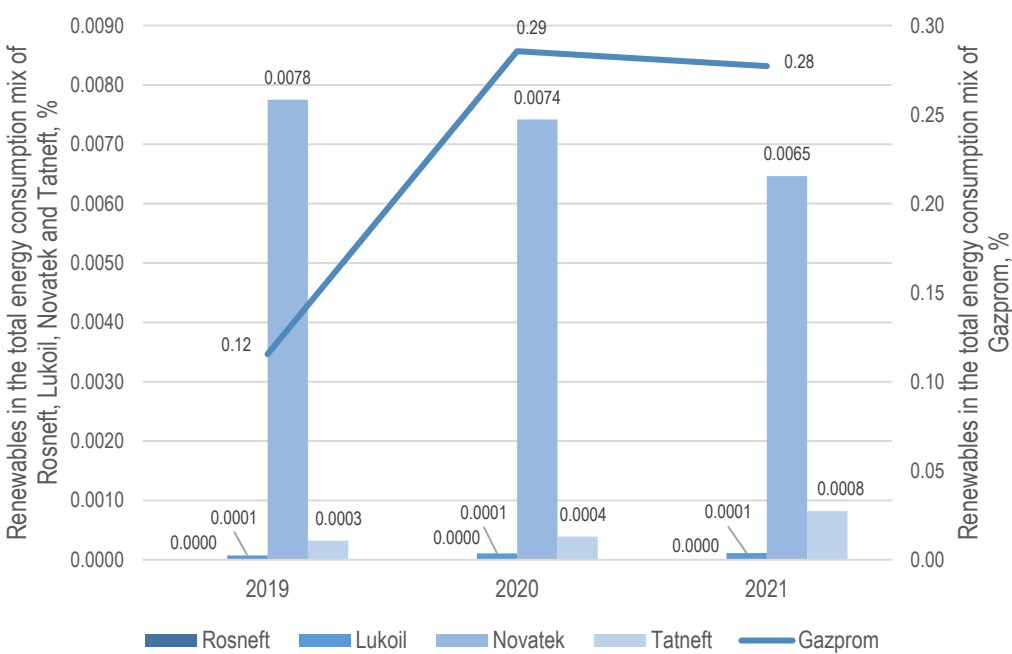

**Figure 2.** Renewables in the total energy consumption mix of Russian O&G companies, %. Source: compiled by the authors based on sustainability reports [50,52,55,57,59,63–72].

As can be seen from Figure 2, Russian O&G companies demonstrate low Target 7.2 contributions: their shares of renewable energy sources in the energy consumption mix do not exceed 1%. Gazprom has the biggest share, which has grown by a factor of 2.33 since 2019. Gazprom utilizes autonomous power sources based on renewable energy in the construction of gas pipelines and production facilities [58].

Next, we will determine how this contribution aligns with corporate strategic goals.

Lukoil is the only company in the sample that sets an ambitious goal of transitioning to 100% renewable energy consumption [53], p. 45. However, the share of renewables has not even reached 1 percent so far, which makes for an insignificant SDG target 7.2 contribution.

The key instruments for ensuring the contribution of Russian O&G companies in this domain are the use of renewable energy sources to power their oil and gas facilities. Currently, the use of solar, wind, geothermal energy, and waste for energy supply in production operations is observed on a minor scale [53,60].

Our analysis of corporate contributions to SDG 7 demonstrates that the total contribution of the Russian O&G sector can only be evaluated based on SDG target 7.2. However,

the data show that this contribution is insignificant. The ambitious goals to expand the use of renewable energy sources (Novatek) and switch to 100% renewable energy consumption (Lukoil) do not align with the current energy consumption mix, as the share of renewables not only fails to reach significant values but also does not increase.

### 3.4.2. Corporate Contributions to SDG 13 (Climate Action)

Indicator 13.2 reflects the annual greenhouse gas emissions, and changes in this indicator can be used to assess the feasibility of achieving the company's strategic goals regarding climate change. In addition, the indicator reflects the corporate contributions to Target 13.2. Although GHG accounting methods are still under discussion, all Russian O&G companies in the final sample maintain the records of emissions for Scopes 1 and 2, disclosing their data in a million tons of $CO_2$ equivalent. They have been keeping records since 2012 (Gazprom and Rosneft), 2013 (Novatek), and 2017 (Lukoil and Tatneft), making it possible to study GHG emission trends over a five-year period. Figure 3 illustrates the changes in total GHG emissions from 2017 to 2021.

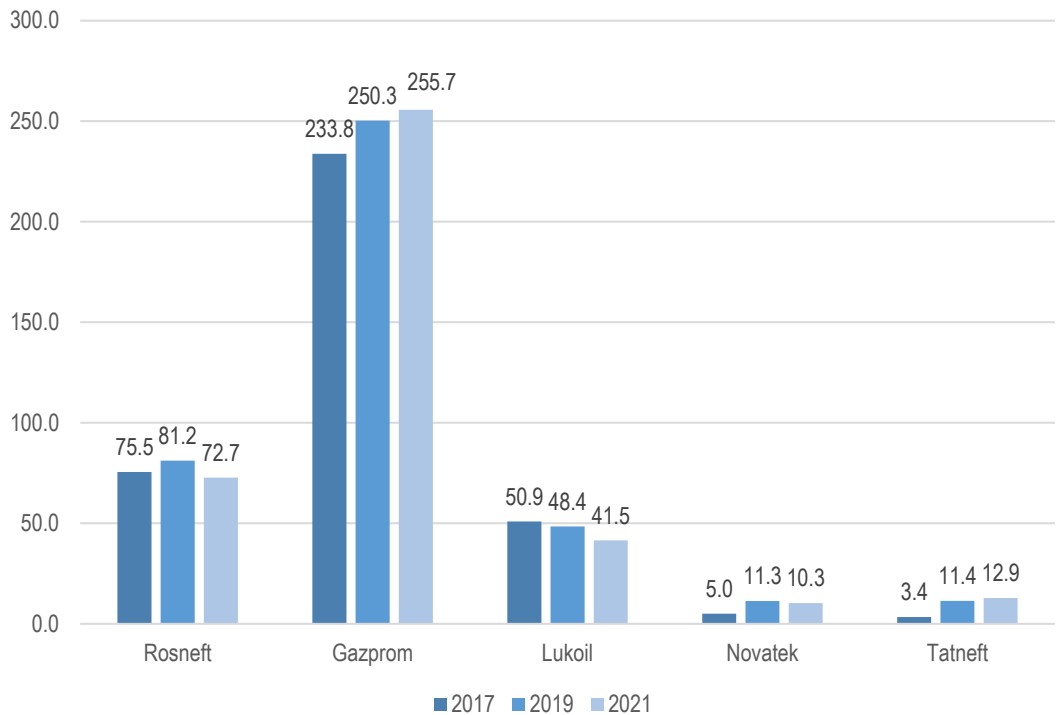

**Figure 3.** GHG emissions, million tons of $CO_2$-eq. Source: compiled by the authors based on sustainability reports [51,53,56,58,60,64–74].

Russian O&G companies increased their emissions by 24.4 million tons of $CO_2$-eq in 2021 compared to 2017, representing a relative increase of 7%. In comparison to 2017, Gazprom's emissions grew by 9.4%, Novatek's emissions doubled, and Tatneft's emissions increased by a factor of 3.8. On the other hand, Rosneft saw a decrease of 2.8 and Lukoil experienced a decrease of 9.4 in emissions over the period. Next, we will determine the extent to which strategic goals align with the actual progress made in terms of Target 13.2 contributions.

Lukoil and Rosneft demonstrate the greatest progress in meeting their strategic goals in decarbonization and climate change. Lukoil's goal of reducing greenhouse gas emissions (Scope 1 + Scope 2) by 20% compared to 2017 has almost been achieved, as the company has already reduced emissions by 18.5%. Rosneft's strategy aims for a 5% reduction in absolute emissions by 2025 compared to 2019, and in 2021, the company reduced its greenhouse gas emissions by 11%, indicating that the goal has been achieved. However, the positive contributions of these individual companies do not offset the negative impact of others.

The current contributions from Gazprom, Novatek, and Tatneft do not provide sufficient grounds to suggest the achievement of their strategic goals, as the greenhouse gas emissions of these companies increased over the five-year period. Therefore, when considering the Russian O&G sector, we cannot claim there is a significant Target 13.2 contribution, as the total GHG emissions grew over the five-year period.

## 4. Discussion

Out of the initial sample of the seven largest O&G companies in Russia, only five companies had up-to-date sustainability reports on their websites, meeting the criterion set in the study. Comparing this result with KP Demirkan's study that showed 86% of companies in the energy sector publish sustainability reports, we can say there is room for improvement for Russian O&G companies in terms of sustainability reporting [75].

Our analysis of SDG prioritization in the Russian O&G sector shows that companies have a clear focus on SD, taking a broad perspective on related issues. This focus aligns with the SDGs selected as priorities, which correspond to the concept of sustainable energy. However, a disadvantage of this focus is the insufficient attention given to other key SDGs, such as SDG 9 (Industry, Innovation, and Infrastructure), SDG 11 (Sustainable Cities and Communities), and SDG 12 (Responsible Consumption and Production). Researchers have defined these goals as crucial for ensuring the SD of the oil and gas sector [38], which means there is a potential for growth in this area.

The investigation demonstrates that the strategic goals of Russian O&G companies formulated for SDG 7 and SDG 13 correspond to UN Targets 7.1, 7.2, 7.3, and 13.2 (Table 4). However, an individual analysis of each company demonstrates that Russian O&G companies do not formulate strategic goals with regard to their contributions to SDG 7.1 related to increasing energy access. This is because this target is not so relevant in Russia, as energy is available and gas and electricity prices are moderate. However, the focus on other SDG 7 targets allows us to identify companies that focus on energy efficiency (Rosneft, Gazprom), as well as on renewable energy sources (Lukoil, Novatek).

Our analysis of the alignment between corporate indicators and contributions to SDG 7 and SDG 13 reveals that the reported results often do not correspond to the UN's targets or indicators. This hinders the assessment of the industry's impact on climate change and the achievement of SDG 7 and SDG 13. Failing to report according to the UN's methodology complicates the energy transition process and the attainment of decarbonization goals. There are several problems associated with improving corporate sustainability reporting.

The first problem is the discrepancy between the indicators used to disclose information in the report and the formulation of the target. In their formulations, SDG targets sometimes signal what indicator should be used (for example, SDG target 7.3: "Double the improvement in energy efficiency"). In all the reports examined, no information was found on a significant increase in energy efficiency. Moreover, the companies do not follow the same vector as to whether to use absolute or relative indicators and units of measurement to reflect their contributions.

The second problem is the mismatch between the indicators used in the sustainability reports and the UN's target indicators. Companies tend to disclose the costs associated with their SDG contributions rather than the actual results achieved. For example, companies report on the amount of investment in renewable energy sources, but they do not disclose their shares of renewable energy consumption, which could serve as their Target 7.2 contributions. Also, they report on the volumes of fuel and energy resources saved both in physical and monetary terms, but they do not include energy intensity indicators in the section dedicated to the SDG contribution. While energy intensity indicators can be found in the annexes to the reports, the units of measurement for these indicators vary among companies.

Our analysis of SDG 7 and SDG 13 contributions made by Russian O&G companies indicates a lack of significant progress in alignment with the indicators defined by the UN's methodology. What supports this statement is that the share of renewables in the total

energy consumption mix of Russian O&G companies did not reach 0.3% over the period from 2019 to 2021, and the total greenhouse gas emissions increased from 2017 to 2021. Still, the companies under study hold quite high positions in terms of ensuring the achievement of the SDGs. This is because Russian O&G companies demonstrate a balanced approach to sustainability based on the TBL concept. This means that sustainable practices focus on the quality of life (social dimension) for SDGs 3 and 4, energy security (social and economic dimensions) for SDG 9, and environmental sustainability (environmental and economic dimensions) for SDGs 7 and 13.

To make a better SDG 7 contribution, Russian O&G companies should promote solar, wind, geothermal energy, and waste as energy sources for their production operations. Additionally, companies can explore opportunities for selling renewable energy.

To make a more substantial SDG 13 contribution, Russian O&G companies need to focus on producing low-carbon hydrogen and advancing technologies for carbon capture, utilization, and storage. Currently, carbon capture, use, and storage projects in the Russian O&G sector are in their early stages. Companies should invest in the development of cogeneration technologies, energy-saving methods for enhanced oil recovery, the optimization of production processes, energy flow distribution methods, and heat exchange methods. It is important to implement tools such as energy management systems, energy efficiency monitoring, cost optimization for lighting, electricity, and heating, as well as the use of energy-saving equipment and technologies.

Target 13.2 contributions can be increased via the implementation of carbon management systems. However, only two out of five companies in the study reported having a well-developed system for monitoring and controlling methane emissions. Therefore, a key recommendation is for companies to establish monitoring systems using instruments both on the ground and in the air. Production equipment upgrades are essential and should be prioritized. Russian O&G companies should also continue exploring the potential of developing new technologies for the utilization of associated petroleum gas (APG) in the future, which forms one of the prerequisites for transitioning to a circular economy.

However, the regional and country-specific nature of Russian O&G companies allows us to identify similarities with the SD models used in Asia and Europe [40]. For instance, Russian O&G companies prioritize compliance with occupational health, safety, and emergency prevention regulations, which is typical for the European SD model. Meanwhile, the focus on achieving SDGs 7 and 13 in Russia aligns with the Asian model, which recognizes the high importance of renewable energy sources, although major projects in this area are yet to be implemented.

At the same time, Russian O&G companies adhere to the international non-financial reporting standards, unlike Asian companies, which is characteristic of the European SD model. Summing up, the Russian SD model can be classified as a hybrid of the Asian, European, and American models.

It is important to note that this study has limitations as it relied solely on sustainability reports as a source of information. The analysis reveals that not all relevant information is consistently presented in these reports. For example, not all Russian O&G companies adequately reflect their contributions to the implementation of SDG 7.1, which aims at achieving universal access to modern energy. The key limitation of the study is its focus on SDGs 7 and 13, even though we understand that O&G companies can greatly influence the achievement of other SDGs.

## 5. Conclusions

As the present study shows, the problem of assessing O&G companies' contributions to the achievement of the SDGs is complicated by several factors. First, not all SD reports disclose enough information. Second, the assessment of the contribution is complicated by the influence of the individual SD strategy of each O&G company, which determines the structure of sustainable practices.

This study highlights the challenges associated with assessing the contribution of O&G companies to the achievement of SDGs. The study reveals that evaluating the total contribution of O&G companies to SD requires comparable data. One key factor is the format of sustainability reporting. The information provided in sustainability reports often does not align with the indicators and targets defined by the UN's methodology. If it does, the unit of measurement may vary among companies. When the indicators reflect relative contributions, they cannot be converted in order to track the contribution of O&G companies within a country or territory. To mitigate the impact of this factor on SDG achievement, it is crucial to continue efforts to standardize reporting indicators in the field of SD. This includes using standard units of measurement for industry indicators and aligning the Global Reporting Initiative (GRI) standards with the indicators defined by the UN's methodology. Therefore, as a recommendation, it is necessary to align the indicators used in the sustainability reports with the UN's methodology in terms of targets and indicators. This will ensure consistency and enable a more accurate analysis of SDG contributions.

The findings of this study will be valuable for O&G companies to enhance their policies and practices in the field of SD. It is recommended that companies update the content of their sustainability reports to provide stakeholders with comprehensive and accurate information.

This study does not only provide recommendations for indicator reporting but also sheds light on the necessary actions to increase the contribution to priority SDGs. Our future research will explore the potential contribution of decarbonization efforts in the O&G sector by adopting circular economy principles, drawing on the international experiences and business models applied in other countries.

**Author Contributions:** Conceptualization, A.C. and N.T.; Methodology, A.C. and T.G.; Supervision, A.C. and T.G.; Writing—review, A.C. and N.T. and editing, T.G.; Modelling and writing original draft, N.T.; Investigation, N.T. All authors have read and agreed to the published version of the manuscript.

**Funding:** This research was funded by the Russian Science Foundation, grant number 22-78-10181, "Decarbonization of the Russian oil and gas complex: conceptual framework, new interfaces, challenges, technological and managerial transformations", https://rscf.ru/project/22-78-10181/ accessed on 30 September 2023).

**Data Availability Statement:** The data are available upon specific request to the authors.

**Conflicts of Interest:** The authors declare no conflict of interest.

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
