# Peer review of "Meeting the UN’s Sustainable Development Goals in the Decarbonization Agenda: A Case of Russian Oil and Gas Companies"

_resources, doi:10.3390/resources12100121_

Round 1

Reviewer 1 Report (Previous Reviewer 3)

Accepted

Accepted

Author Response

Dear Reviewer,

Thank you for appreciating our work

Reviewer 2 Report (New Reviewer)

I would like to thank the authors for this scientific contribution.

Few tips: the literature section, as well as the methodological section, should cite authors that have done the same work (even if in different sectors). 

I suggest to read papers of "Rosati and Faria" and "Annesi and Battaglia". Following, few titles: 

Rosati, F., & Faria, L. G. (2019). Addressing the SDGs in sustainability reports: The relationship with institutional factors. Journal of cleaner production, 215, 1312-1326.

Rosati, F., & Faria, L. G. D. (2019). Business contribution to the Sustainable Development Agenda: Organizational factors related to early adoption of SDG reporting. Corporate Social Responsibility and Environmental Management, 26(3), 588-597.

Battaglia, M., Gragnani, P., & Annesi, N. (2020). Moving businesses toward sustainable development goals (SDGs): evidence from an Italian “benefit-For-Nature” corporation. Entrepreneurship research journal, 10(4), 20190305.

Battaglia, M., Annesi, N., Calabrese, M., & Frey, M. (2020). Do agenda 2030 and Sustainable Development Goals act at local and operational levels? Evidence from a case study in a large energy company in Italy. Business Strategy & Development, 3(4), 603-614.

Results/Discussion should highlight that the limits of the study are also related to the limited focus on the SDGs 7 and 13, while the sector can have larger potential also on the others SDGs. 

Author Response

Dear Reviewer, thank you for your comments and appreciation of our efforts. Detailed responses to your comments are provided below

Comment: Few tips: the literature section, as well as the methodological section, should cite authors that have done the same work (even if in different sectors).

I suggest to read papers of "Rosati and Faria" and "Annesi and Battaglia". Following, few titles: 

Rosati, F., & Faria, L. G. (2019). Addressing the SDGs in sustainability reports: The relationship with institutional factors. Journal of cleaner production, 215, 1312-1326.

Rosati, F., & Faria, L. G. D. (2019). Business contribution to the Sustainable Development Agenda: Organizational factors related to early adoption of SDG reporting. Corporate Social Responsibility and Environmental Management, 26(3), 588-597.

Battaglia, M., Gragnani, P., & Annesi, N. (2020). Moving businesses toward sustainable development goals (SDGs): evidence from an Italian “benefit-For-Nature” corporation. Entrepreneurship research journal, 10(4), 20190305.

Battaglia, M., Annesi, N., Calabrese, M., & Frey, M. (2020). Do agenda 2030 and Sustainable Development Goals act at local and operational levels? Evidence from a case study in a large energy company in Italy. Business Strategy & Development, 3(4), 603-614”.

Answer: Thank you for your recommendation for expanding the literature review. It was really difficult for us to find articles similar to the objectives of our study. All the articles you mentioned are very valuable in the context of our research. We wrote a separate paragraph on the article " Do agenda 2030 and Sustainable Development Goals act at local and operational levels? Evidence from a case study in a large energy company in Italy, by Rosati, F., & Faria, L. G. D. " because we felt that it was closest to our research objectives. Please look at the paragraph highlighted in yellow color, page 3.

We also mentioned the other articles in the literature review. They are highlighted in yellow. Thank you 

Comment: “Results/Discussion should highlight that the limits of the study are also related to the limited focus on the SDGs 7 and 13, while the sector can have larger potential also on the others SDGs”. 

Thank you very much for this comment. You are absolutely right. We have added this limitation at the end of the discussion section (in yellow).

Reviewer 3 Report (New Reviewer)

The paper is a brief overview of Russian company reports to draw conclusion about the sustainable efforts in the country.  Very little is reported and this means very little is done.  Perhaps the Russian company reports should be compared to the approaches by companies in the US, Europe, Middle East and Asia to see how the Russian reports compare.  

There is no mention of gas flaring in this paper and yet the World Bank identifies Russia as the #1 country in terms of flaring of gas.  

See

https://www.worldbank.org/en/programs/gasflaringreduction/global-flaring-data

I am not sure how you can reach any conclusions from 6 page reports on climate sustainability when there are billions being invested in energy.  

There needs to be editing.  The paragraphs beginning 'A study by F. M. M. G' are duplicated.

There are extraneous words.  The number of the references is off.  Perhaps this will be handled in production.

Author Response

Dear Reviewer, thank you very much for the feedback on our paper. We sincerely thank you for your time and valuable recommendations. We have checked all your remarks. Please, see below the detailed answers

Comment: “The paper is a brief overview of Russian company reports to draw conclusion about the sustainable efforts in the country.  Very little is reported and this means very little is done. Perhaps the Russian company reports should be compared to the approaches by companies in the US, Europe, Middle East and Asia to see how the Russian reports compare.”  

Answer: Thank you for your comment. We understand your points. But let us notice that Russia is on the way to sustainable development. At the same time, open reporting has been developing in Russia for about 15 years. Leading companies prepare their reports in accordance with the key principles of the Global Reporting Initiative (GRI) standards, though in some cases more attention is paid to the fact of openness than to the real figures. Perhaps this could be interpreted as little environmental and social responsibility, but it is not true. Much is done both in terms of the environmental performance enhancement and social initiatives (especially at the regional level). Major export-oriented companies follow approaches typical of their competitors functioning across the world.  

However, we agree with you that not much is done on climate, that’s why our research is focused on the real contribution of O&G companies to SDGs achievement in the terms of decarbonization. Our research has revealed that the total GHG emissions grew over the five-year period, and the share of renewables in energy consumption is low. That’s why there is no significant Target 13.2 and Target 7.2 contribution from Russian O&G companies. We have restructured the discussion part of the paper to emphasize more clearly that the contribution of Russian oil and gas companies is rather low (please see the revised Discussion and especially the paragraph on page 20 (in green). However, the assessment of this contribution is complicated by the peculiarities of corporate reporting, which we have also tried to reflect in the discussion and conclusion. Please take a look at the new version of the discussion and conclusion (in gray).

As for the comparing Russian company reports with the companies in the US, Europe, Middle East and Asia, it was out of the purpose of our research. This is due to the fact that we have not found studies that analyze the real contribution to SD made by Russian O&G companies in the context of global climate change. We tried to fill this gap. But thank you for this idea, it would be interesting. We will think about it in our further research.

Comment: “There is no mention of gas flaring in this paper and yet the World Bank identifies Russia as the #1 country in terms of flaring of gas.  

See

https://www.worldbank.org/en/programs/gasflaringreduction/global-flaring-data”

Answer: Thank you for your comment. We agree with you that this is highly important issue for O&G industry in the climate agenda. As a result of our previous studies (Utilization of associated petroleum gas in Russia: methods and prospects for the production of gas chemistry products, please see http://iep.kolasc.net.ru/journal/eng/?page_id=902), we have thoroughly investigated the problem, resulting in the following conclusions. The analysis of the level of APG utilization by the largest Russian oil and gas companies showed that companies are already actively engaged in this problem by developing gas chemical activities, however, they often use simpler methods of utilization (injection of APG into the reservoir, generation of electricity, etc). Leading companies (those which are described in the article) do report that they utilize up to 90-99% of APG (Lukoil, Tatneft and Novatek - 96%, Gazprom - 90% in 2021). Since open reports are documents audited by the third parties, there is no reason not to trust this information. Still, in this article, we don't discuss performance of the smaller companies which can have high level of gas flaring, but it was out of our research.

However, let us notice that the purpose of our article is to analyze the contribution using the UN methodology. In the UN indicators the volume of gas flaring is not used.

Comment: “I am not sure how you can reach any conclusions from 6 page reports on climate sustainability when there are billions being invested in energy.  

Answer: Thank you for your comment. Let us clarify that the base of our research consists of SD reports, which are 70-100 page documents, prepared in accordance with recommendations of GRI standards. For example, please look at the link below:

Comment: “There needs to be editing.  The paragraphs beginning 'A study by F. M. M. G' are duplicated”.

Thank you for your comment and attentiveness to our article. We have corrected our mistake.

Reviewer 4 Report (New Reviewer)

The presented paper was difficult to understand due to its dense layout, text in black and red color intertwined, and multiple occurrences of the same text (e.g. lines 116 and 159).

Being an engineer, I expected more data in tables and graphs, and less focus on methodological issues with corporate reporting or conforming to the same parameters and their units.

It seems like an initial raw draft is being presented here instead of being discussed and scrutinized among the authors' colleagues and supervisors.

The flow of the discussion is fine.

I would recommend to resubmit this work with a focus on clarity, data presentation and itemized conclusion.

English is fine given the formatting issues are fixed.

Author Response

Dear Reviewer, thank you very much for the feedback on our paper. We sincerely thank you for your time and efforts. We have checked all your remarks and revise the paper in an appropriate way.

Comment: “The presented paper was difficult to understand due to its dense layout, text in black and red color intertwined, and multiple occurrences of the same text (e.g. lines 116 and 159)”.

Answer: Thank you, we understand. We have corrected the file that is uploaded to the publisher to a version that does not show the corrections. We apologize for the inconvenience. Please take a look at the new version of this article. We hope you will find it satisfactory.

Comment: “Being an engineer, I expected more data in tables and graphs, and less focus on methodological issues with corporate reporting or conforming to the same parameters and their units”.

Answer: Thank you for your valuable comment. We understand your point of view, but we would like to emphasize that our publication focuses on managerial issues, and methodological issues are really important in this context. This is due to the fact that approaches to assessing the contribution to sustainable development are a debatable issue, and we have not found any analogs of such a study on the example of the Russian oil and gas complex.

The purpose of the article is not to reveal engineering solutions of sustainable initiatives in the context of global climate change, but is of managerial nature and is aimed at analyzing the real contribution, assessing the compliance of the corporate indicators of the Russian oil and gas companies used for monitoring the contribution to sustainable development with the indicators approved by the UN. In our article we used open data as much as possible.

In our study we have addressed, among other things, the issue of using unified parameters and their units in corporate reporting. We emphasize that, one way or another, how close the energy sector is to achieving the SDGs is largely determined by the activities of companies in the oil and gas sector. The requirement to disclose information in the same units of measurement is not spelled out in standards such as GRI. Keeping records and disclosing information in such specific indicators as energy intensity for different indicators makes it difficult to assess progress towards achieving the SDGs at the level of the entire industry of a particular country, as well as on a global scale.

In any case, the assessment of the contribution to the achievement of the SDGs is determined by the factor of information disclosure in corporate reporting, which should meet the requirements of completeness, reliability and transparency.

Comment: “It seems like an initial raw draft is being presented here instead of being discussed and scrutinized among the authors' colleagues and supervisors.

The flow of the discussion is fine.

I would recommend to resubmit this work with a focus on clarity, data presentation and itemized conclusion.

Thank you very much for your valuable recommendations. We have totally revised the discussion and conclusion sections and tried to make them itemized (in gray). We hope that the article has improved and that you will be interested in it.

Round 2

Reviewer 4 Report (New Reviewer)

Thank you for the updated version of your paper, having my questions answered helped a lot. However, I still see the paragraph 118-129 duplicated at 167-176?! Is it intentional?

Besides this minor fix, I deem the paper publishable. Congratulations

no issues here

Author Response

Dear Reviewer,

Thank you very much for your appreciation of our work

We have corrected duplicated paragraph

Thanks again 

This manuscript is a resubmission of an earlier submission. The following is a list of the peer review reports and author responses from that submission.

Round 1

Reviewer 1 Report

This study provides a content analysis of sustainability reports of five major oil and gas producing companies in Russia for the period of 2021.

The overall aim is to determine how well the sustainability goals and strategies of these O&G companies align with the UN Sustainable Development Goals.

In its current form, the methodology is not systematic enough and therefore the analysis is twoo weak to draw sufficient conclusions on the sustainability practices and future sustainability strategies of these Russian O&G companies.  

Indeed, the authors themselves note that “O&G companies around the world tend to choose and implement practices that can generate quick results and appeal to shareholders in their reports. Consequently, companies may disclose sustainable practices in their reports that do not make a significant contribution to the achievement of sustainable development 153 goals and targets.”

To make a reasonable assessment of Russian O&G industry’s commitment to sustainability, the reader must be able to understand, firstly – what are best practice standards for ESG and Sustainability Reporting and how well do the reports analysed in this study fit these criteria (e.g. level of transparency, etc)? And secondly – what are the performance indicators for each of the SDGs and what evidence have these companies provided to show they have taken, are taking or plan to take action that will meet these indicators, goals and targets?  

Some suggestions are provided to improve the quality of the manuscript:

Introduction

1. The authors raise the issue of “coherence between national and global SDGs. O&G companies may focus their sustainability strategies on contributing to meeting the targets of national government strategies related to environmental, social, and economic development.” The authors should briefly mention how Russian SDGs compare with global SDGs and where key differences lie.

2. The authors talk about addressing the gap of sustainability reporting for managerial operations but they need to provide further explanation of what this means or looks like, what criteria is assessed and how they establish whether the reports are being used to inform managers.

Methods:

3. GRI Reporting Standards Framework: The authors cite the GRI Standards as that most commonly applied for Sustainability Reporting. Thus they should explicity outline the key elements of the GRI Standards as part of the methodology and use this as a guiding framework to systematically assess the quality of the five sustainability reports. GRI Standard 11 is also specific for the Oil & Gas Industry – GRI standards are easily downloadable from their website: https://www.globalreporting.org/how-to-use-the-gri-standards/gri-standards-english-language/ 

4. SDG Performance Indicators: the authors should show what the performance indicators are for each of the SDG targets. The most obvious indicator for the O&G industry is: 13.2.2 Total greenhouse gas emissions per year. Is there evidence to show that O&G companies GHGs are declining? If they have GHG reductions targets, how do they plan to achieve these? Are the plans feasible? Are the targets in line with SDG goals? https://unstats.un.org/sdgs/indicators/Global%20Indicator%20Framework%20after%202022%20refinement_Eng.pdf 

Results: 

5. The results are presented unsystematically and require further evidence to back up statements and claims. Particularly in the tables of section 3.3, they should expand to include direct quotes from each report for each company. 

E.g. A table or results should include the following columns or presentation of data:

[SDG Target]. Example: 13.1. Strengthen resilience and adaptive capacity to climate-related disasters

[SDG Indicator]. Example: 13.2.2 Total greenhouse gas emissions per year.

[Company]: Example: Rosneft

[Company Targets]: Example: Rosneft’s 2021 ‘Sustainable Energy’ Report (p50) states its targets include: 

o Short-term Reduction of absolute GHG emissions of Scope 1 and 2 by 5% by 2025

o Medium reduction of absolute GHG emissions of Scope 1 and 2 by over 25% by 2035

o reduction of methane emissions intensity to below 0.2% by 2030

o zero routine flaring of APG by 2030

o reduction of unit GHG emissions of Scope 1 and 2 in exploration and production to below 20 kg of CO2  equiv. per boe by 2030 or sooner

o Long-term Scope 1 and 2 carbon neutrality by 2050

[Key Levers / Actions to achieve target]:

[Progress to date]:  Example: In 2021, Rosneft stated it had achieved energy savings amounted to 372 thousand tonnes of reference fuel or 10.9 mln GJ, which is 49% above the plan.

Reviewer 2 Report

It is an interesting topic, however, current version makes an impression of auditing and reviewing a few reports, rather than a research. There is no methodology and no findings per se, apart from stating that in 2021 five O&G companies published their sustainability reports with the inclusion of different SDGs. This is not sufficient for a research article, but perhaps could be improved to a level of a review paper.

A few more specific comments:

1) The title is misleading, should be corrected. There is no analysis of "achieving the UN's SDs" at all, just auditing a few reports in a single year. Alternatively, try to include some analysis over time. Why focus on a single year, and how can we measure an improvement in this case?

2) Introduction could be made shorter and should be focused on the topic. Some parts at the end can be included in the methodology. In the introduction, try to avoid limiting sustainable development concept to UN's SDGs, these are not the same.

3) Methodology needs to be fully rewritten. How can others to use and apply your approach elsewhere? What are the main steps in your research (note, these are not the steps in writing your paper)? As a suggestion, on the diagram, clearly show how the companies' screening has been done, how many companies in total in every screening step, what are the parameters, time period, and main research/analysis steps.

4) It would be great to indicate whether/if/how your final sample is representative. E.g. per cent of total number of companies, total revenue in the sector, employment, etc. Are they all large multinational companies, or mid-tier, etc.?

5) The results section needs significant improvement, and should be structured in the same way as described in the methodology. Some tables/graphs can go into Supplementary materials. Try to focus on SDGs achievements and real-life examples, not just counting how many times different SDGs have been mentioned in the reports.

6) There is an interesting Figure at the end of the Discussion section, but not clear why do you need this and whether this has been used at all. Perhaps, introduce much earlier and explain how to use this, otherwise consider to remove.

7) Try to avoid referring to SDGs numbers, instead refer to the actual issue and keep SDG number in brackets, e.g. "(SDG 13)". Not everyone knows/remembers SDGs by heart.

8) Conclusion needs improvement, and needs to be clear. What has been achieved? E.g. new methodology developed and applied, and/or analysis of 2021 O&G companies reporting. What are the major findings? Why and how can you recommend anything based on a single year reporting of a few companies?

Reviewer 3 Report

Attached
